# Workplace Mistreatment and Health Conditions Prior and during the COVID-19 in South Korea: A Cross-Sectional Study

**DOI:** 10.3390/ijerph192012992

**Published:** 2022-10-11

**Authors:** Nataliya Nerobkova, Soo Young Kim, Eun-Cheol Park, Jaeyong Shin

**Affiliations:** 1Department of Public Health, Graduate School, Yonsei University, Seoul 03722, Korea; 2Institute of Health Services Research, Yonsei University, Seoul 03722, Korea; 3Department of Preventive Medicine, Yonsei University College of Medicine, Seoul 03722, Korea; 4Department of Policy Analysis and Management, College of Human Ecology, Cornell University, Ithaca, NY 14850, USA

**Keywords:** health conditions, working condition survey, workplace mistreatment, workplace abuse, workplace discrimination

## Abstract

Background: This study examined the relationship between workplace mistreatment, including discrimination, abuse, and overworking, and health problems among full-time workers prior to and during the coronavirus disease (COVID-19) outbreak in South Korea. Methods: We analyzed data from the 2017 and 2020–2021 Korean Working Conditions Surveys, including the final sample of 44,425 participants. Multiple logistic regression was used to examine the relationship between workplace mistreatment and health problems among workers by gender. Interaction analysis was conducted to establish the association between the COVID-19 pandemic and health problems related to mistreatment. The occupational, demographic, and socioeconomic backgrounds were adjusted. Results: We found a significant association between workplace mistreatment and health problems, including headaches, eyestrain, and anxiety. The association increased after the COVID-19 pandemic: “discrimination” (men (OR 2.26, 95% CI 1.93–2.65), women (OR 2.73, 95% CI 2.36–3.17)); abuse (men (OR 5.42, 95% CI 2.87–10.23), women (OR 4.70, 95% CI 3.12–7.08)); and overworking: men (OR 2.36, 95% CI 2.01–2.77), women (OR 3.52, 95% CI 2.68–4.61). The interaction indicates an increased incidence of people having health problems due to workplace mistreatment (OR 1.03, 95% CI 1.00–1.06) during the COVID-19 pandemic. Conclusion: Statistically, employees who experience workplace mistreatment have worse health conditions. The COVID-19 pandemic has affected the job environment and increased the association between workplace mistreatment and health problems. To eliminate the health problems related to workplace mistreatment, it is necessary to address the impact of the COVID-19 pandemic on work and employee health conditions.

## 1. Introduction

Workplace mistreatment has been recognized as a tremendous public concern within work and social environments [1,2]. Workplaces are environments for both economic and social engagement. Thus, some employees are inevitably exposed to workplace mistreatment, resulting in occupation-related health conditions. It may affect work engagement and lead to higher emotional exhaustion, burnout, low self-esteem, poor physical and mental health, and personal well-being [3]. The current research was designed to examine the working environment prior to and during the COVID-19 pandemic and identify to what extent the full-time workers in Korea operating in such settings were exposed to workplace mistreatment and associated health problems.

The estimated prevalence varies due to the lack of a consistent definition and perception of workplace mistreatment. The prevalence of workplace mistreatment also varies because studies differ in terms of the timeframe and frequency of exposure [4]. Therefore, our study broadly defines “workplace mistreatment” as the aspects of a working environment that may negatively impact workers’ health and well-being, including discrimination, physical abuse, sexual harassment, bullying/harassment, and overworking.

Workplace discrimination refers to all means of expressing and institutionalizing social relationships of dominance and oppression. Research has suggested that pervasive work-related discrimination based on the gender and educational level of workers exists in South Korea [5]. Discrimination based on hire form and income was significantly associated with poor self-perceived health status among employees [5]. However, this study lacked consideration of workers’ experience of harassment and violence.

Abuse can be defined as any form of aggressive behavior aimed at harming another human being. Workplace violence is one of the most compound and hazardous occupational risk factors related to social relationships leading to health problems in a work environment [6], and its health effects are relatively consistent. Experiencing workplace violence, bullying, sexual harassment, and verbal violence increases the risk of mental health problems [7].

Overwork-related disorders, such as cerebrovascular/cardiovascular diseases (CCVDs) [8] and mental disorders due to overwork [9], are also significant occupational and public health issues worldwide, particularly in East Asian countries [10]. Some studies in South Korea have found that long working hours and high work intensity were associated with CCVDs [11,12]. Other studies [13,14] on the relationship between suicide and overwork in Korea indicated that overworking may be a risk factor for depressive symptoms or suicidal ideation.

To date, little is known about the changes in the working environment due to the COVID-19 pandemic’s impact on workplace mistreatment and associated health conditions. However, some studies examined the association between COVID-19 and changes in working environments. Therefore, the longitudinal study in Korea established that customer incivility significantly impacted workers’ performance during the pandemic [15]. Another mixed-methods study that explored the impact of the pandemic on workplace violence at an academic emergency department emphasized a positive association of incidents of workplace violence with the COVID-19 case rate [16]. The results of a cross-sectional study on the nurses working during the pandemic similarly showed that a significant proportion of nurses who cared for patients with COVID-19 experienced more physical violence and verbal abuse [17]. Several studies also emphasized the increased workplace discrimination risk during the pandemic [18,19,20]. Nevertheless, the mentioned studies mainly focused on the general association between the pandemic and working conditions without considering its impact on workers’ health.

For that reason, the objective of this study is to examine the research hypothesis: if there are any differences in the association between workplace mistreatment and health problems prior to and during the COVID-19 pandemic. A clear understanding of this relationship will provide substantial implications to policymakers and employers for considering how to best support a positive working environment and prevent adverse health outcomes in workers. To test our hypothesis and fill gaps in the literature, we attempted to evaluate three leading elements of workplace mistreatment: discrimination, abuse, and overworking, as well as the risk for various health problems among Korean full-time workers before and during the COVID-19 pandemic.

## 2. Materials and Methods

### 2.1. Data Source and Sample

We obtained the data for the current cross-sectional study from the fifth (2017) and sixth (2020–2021) Korean Working Conditions Survey (KWCS) conducted by the Research Institute Occupational Safety & Health Research Institute (OSHRI). The data were collected using an electronic questionnaire installed on a tablet PC between July and November 2017 and between October 2020 and April 2021 in 17 cities and provinces. The survey investigated the working environment and health conditions of Korean workers. The KWCS has benchmarked the European Working Condition Survey research contents and methods [21] and uses a complex sampling design, including rigid multi-stage stratified cluster random sampling and weighted values.

### 2.2. Participants

The population of the KWCS includes all employees aged over 15 years in 50,000 households (one person per household) at the time of the survey. The survey was conducted by choosing participants considered as “employed persons” within households selected in the sample group.

The total number of participants surveyed was 50,205 in 2017 and 50,538 in 2020–2021. The exclusion criteria included individuals aged 19 years or younger and data with missing information. Data on 44,425 participants (21,460 from 2017; 22,965 from 2020–2021) were selected; full-time workers comprised the representative sample. Because the OSHRI is a secondary dataset available for public use without identifiable information, our study did not require approval from the institutional review board.

### 2.3. Variables

“Workplace mistreatment” was our variable of interest, assessed by three questions: (1) “Over the past 12 months, have you experienced any discrimination at work?”—discrimination (yes/no); (2) “Over the past 12 months, during the course of your work, have you been subjected to any of the following: physical violence, sexual harassment, bullying/harassment?”—abuse (yes/no); and (3) “In the past month, has it happened at least once that you had less than 11 h between the end of one working day and the start of the next working day?”—overworking (yes/no). For the primary analyses, we considered each mistreatment type as an individual factor for our outcome variable. For the subgroup analysis, we also examined the association with “sexual harassment” in women specifically. Furthermore, the interaction analysis included the comparison of workers who experienced mistreatment (workers who answered “no” to all the questions were classified as having no experience of workplace mistreatment; those who responded “yes” to at least one of the questions were classified as workers with experience of workplace mistreatment), and year (2017 and 2020–2021).

Our outcome variable, “health problems,” was evaluated with the question, “Over the past 12 months, did you have any of the following health problems?” Participants answered “yes” or “no” regarding a number of symptoms, such as “backache,” “muscular pain,” “headache,” “anxiety,” and “fatigue.” Those who responded “no” to all symptoms were classified as having “no health problems,” whereas those who answered “yes” to at least one symptom were classified as having “health problems.” For the analysis, we also considered health symptoms individually: (1) headaches (yes/no), (2) anxiety (yes/no), (3) overall fatigue (yes/no), and (4) physical pain (yes/no) that included backache and muscle pain.

The study includes three working characteristic variables: working hours per week (less than 40 h, 40 h, and more than 40 h), commuting time (less than 30 min, 30 to 60 min, more than 60 min), and monthly income amount (less than 2,000,000 Korean won (KRW), 2,000,000 to 3,000,000 KRW, more than 3,000,000 KRW). Further, 1292 KRW is equivalent to USD 1 (as of 15 June 2022).

We have also included the following sociodemographic variables: gender (men, women), age (20–29 years old, 30–39 years old, 40–49 years old, 50–59 years old, and 60 years old or over), and educational background (high school or less, and undergraduate or more). The model of all employed variables is depicted in Figure 1.

### 2.4. Statistical Analysis

A chi-squared test was conducted to investigate the association between workplace mistreatment and health problems among full-time workers. Due to traditional differences in the dominance of men and women in a working environment as well as the statistical prevalence of sexual harassment towards women [22,23], all analyses included gender to examine gender-specific differences between mistreatment types and the risk of health problems. Multiple logistic regression analysis was performed to evaluate this relationship. The interaction analysis was performed to establish the association between certain periods (before and after the COVID-19 pandemic) and workplace mistreatment with health problems. The results were reported using odds ratios (ORs) and 95% confidence intervals (CIs). Model fitting was performed using the PROC SURVEYLOGISTIC procedure and application of weight. Differences were considered statistically significant with a *p*-value of <0.05. All data analyses were conducted using the SAS 9.4 software (version 9.4; SAS Institute Inc., Cary, NC, USA).

## 3. Results

Table 1 summarizes the participants’ general characteristics. We found that 7728 (36.0%) participants in 2017 and 11,069 (48.2%) participants in 2020 had health problems. Among them, 2012 (51.3%) participants in 2017 and 1880 (67.8%) participants in 2020–2021 experienced workplace mistreatment at some point. Notably, despite an overall decrease in the number of participants who experienced mistreatment (3920 in 2017; 2771 in 2020–2021), there was a significant increase in the percentage of those with health problems (51.3% in 2017; 67.8% in 2020–2021).

Table 2 illustrates the findings of the logistic regression analysis stratified by gender and the association between three types of workplace mistreatment and health symptoms after adjustment to age, year, educational level, income amount/month, working hours/week, and commuting time. The detailed adjustment process for all models is depicted in Appendix A. These results suggested an association between all types of mistreatment and health problems among participants of both genders. The association was consistent throughout the models with such health symptoms as headache, eyestrains (“discrimination”: men (OR 2.40, 95% CI 2.16–2.67), women (OR 2.51, 95% CI 2.27–2.77), abuse: men (OR 3.38, 95% CI 2.35–4.85), women (OR 3.06, 95% CI 2.42–3.85), and overworking: men (OR 2.40, 95% CI 2.15–2.68), women (OR 2.15, 95% CI 1.89–2.52)) and anxiety (“discrimination”: men (OR 2.83, 95% CI 2.09–3.83), women (OR 2.38, 95% CI 1.74–3.26), abuse: men (OR 8.28, 95% CI 4.50–15.23), women (OR 4.29, 95% CI 2.42–7.60), and overworking: men (OR 2.45, 95% CI 1.77–3.37), women (OR 3.02, 95% CI 1.96–4.65)).

The independent subgroup analysis findings of the factors associated with workplace mistreatment type and having at least one of all health problems are shown in Figure 2. Notably, compared to 2017, the magnitude of the association was more significant in 2020–2021 within all mistreatment types: “discrimination” (men (OR 2.26, 95% CI 1.93–2.65), women (OR 2.73, 95% CI 2.36–3.17)); abuse (men (OR 5.42, 95% CI 2.87–10.23), women (OR 4.70, 95% CI 3.12–7.08)); and overworking: men (OR 2.36, 95% CI 2.01–2.77), women (OR 3.52, 95% CI 2.68–4.61). Furthermore, discrimination revealed that women over 59 years of age were the most vulnerable group (OR 3.01, 95% CI 2.09–4.33), while in men, it was the age group of 20–29 years old (OR 2.84, 95% CI 2.21–3.65). Overworking was an issue in all age groups and among both genders. 

Table 3 shows the results of a subgroup analysis of the association between sexual harassment and health problems in women stratified by year. Thus, compared to 2017, in 2020–2021, an increase in the likelihood of having health problems among women exposed to sexual harassment was found (2017: OR 2.09, 95% CI 1.29–3.40; 2020: OR 2.59, 95% CI 1.24–5.39).

Table 4 displays the results of interaction factors associated with health problems in 2017 and 2020–2021. Our results showed a synergy effect, as the effect of “year” on the outcome variable (“health problems”) depended on our variable of interest (“workplace mistreatment”). The interaction between “participants without mistreatment in 2017” with “participants with mistreatment in 2020–2021” indicates that in 2020–2021, the incidence of people having health problems associated with workplace mistreatment increased slightly (OR 1.03, 95% CI 1.00–1.06) (P-for-interaction is 0.0321).

## 4. Discussion

Workplace mistreatment is a significant risk factor for health deterioration. The COVID-19 pandemic placed restrictions and burdens on many population segments, including the working force, which resulted in declining interpersonal relationships [24,25] and changes in the workplace environment. Thus, we investigated the association between certain types of workplace mistreatment and the development of health problems among full-time employees in South Korea before and during the COVID-19 pandemic. Through the analysis of nationally representative survey data, we found evidence to support the existence of this relationship.

The primary outcomes of our study suggested that workers who experienced workplace mistreatment were more likely to have health problems than those who did not. Significant associations were found between each type of workplace mistreatment and health symptoms. Notably, according to the results of the subgroup and interaction analyses, these associations were more prominent during the COVID-19 pandemic in 2020–2021. A cross-sectional study from Japan emphasized similar results with a high prevalence of workplace harassment related to COVID-19, which was associated with poor mental and physical health [26].

Further, compared with other age groups, women over 59 years of age showed the strongest correlation between discrimination and health problems. This association was the most prominent in men in the 20–29 age group. Therefore, our results suggested that these groups were the most vulnerable, supporting that younger and older professionals are more exposed to all forms of prejudice and discrimination [27]. Whereas older female workers face age stereotypes about personality, skills, and adaptability [28], younger male workers might be considered untrustworthy, less responsible, and lacking experience [29]. Our findings also indicated that the association between mistreatment and health problems was more significant in those with higher educational backgrounds. The possible explanation suggests that professionals with a higher educational level, who experienced mistreatment at work, are more likely to consider that event distressing and unacceptable than workers with lower educational backgrounds [1]. Workplace mistreatment, regarded as a low reward, may produce tremendous emotional distress [30,31]. The imbalance between high effort in education and work and low reward would stimulate psychological stress, leading to somatic symptoms [1,32].

Another noteworthy finding requiring further attention was that sexual harassment of women remained a significant issue in the workplace. During the COVID-19 pandemic, the likelihood of women experiencing a co-occurrence of sexual harassment and health problems increased remarkably. Previous studies [33,34] that examined the health effects of age and gender-based mistreatment emphasized that women’s perceptions of sexual harassment and gender discrimination were associated with increased psychological distress. Other studies [35,36] have examined the impact of sexism on physical health. Thus, perceptions of sexual harassment adversely affect multiple dimensions of women’s health.

Due to the COVID-19 pandemic, workers became more exposed to workplace mistreatment, especially regarding harassment or overworking. South Korea experienced three waves of the COVID-19 pandemic since the emergence of severe acute respiratory syndrome coronavirus 2 (SARS-CoV-2) in December 2019 [37]. The COVID-19 pandemic presents a significant challenge to healthcare systems and the working population. Social distancing measures to hamper the spread of the virus, such as working from home and school closures, have placed a tremendous burden [38] on all segments of the population. Although the second and third waves in South Korea began with increasing social activity and poor social distancing, the spread was successfully mitigated by the rapid strengthening of social distancing measures. Such measures included the shift to working from home [39] via telecommunication tools such as email, phone, chat, and video apps. Whether working in physical workplaces or the apparent safety of home, researchers [40] have suggested that women employees continue to be chased by sexual harassment. Thus, sexual harassment has taken the new form of a cyber avatar [40] and remains unnoticed due to the protections afforded by ambiguity.

While overworking is a well-known risk factor for CCVDs, the results are consistent with the data that fundamental working condition changes and occupational health approaches are essential since it affects all groups of workers [41]. Although the National Assembly passed legislation in February 2018 to shorten work hours, with the revised law to be applied on an annual basis depending on the size of the workplace [41], more data is needed to evaluate the effectiveness of this policy. As the COVID-19 pandemic resulted in social restrictions and heavier workloads on the workforce, many researchers [42,43,44] have suggested that the increased workload among health professionals resulted in health problems such as increased levels of stress, depression, anxiety, physical exhaustion, as well as increased rates of CCVDs. While it should not be ignored that health professionals are at increased risk of infection with COVID-19, it is also essential to recognize that professionals in other fields have also been overworked during the COVID-19 pandemic, emphasizing that teleworking leads to constant overworking [45].

The etiology of the association between the pandemic’s impact on workplace mistreatment and health-related conditions has not been fully established, although a possible explanatory mechanism may be suggested. The main factor linking exposure to workplace mistreatment and health outcomes is stress, and workplace mistreatment can be considered a significant job stress factor [1]. The pandemic is a major global challenge, and workers have been forced to change their habits and lifestyle at numerous levels. Work arrangements and conditions have changed considerably, raising new challenges for the well-being and health of workers and evoking greater occupational stress. Moreover, while peer support was hypothesized to be the best tool for preventing and mitigating work stress [46], its contrast, workplace mistreatment leads to greater occupational stress.

The current study has several limitations. First, it was based on data from a cross-sectional study. Therefore, although associations were found between variables, the causal order could not be determined. Second, our study relied on self-reported data. Hence, people may not have accurately reported their health conditions or information about experienced workplace mistreatment. Third, as the cut-off points for workplace mistreatment were adopted from the OSHRI, it may be difficult to generalize our findings to various settings of populations. Moreover, the variables of workplace mistreatment were assessed by a simple set of questions, which lacked detailed information. A more specific set of questions on severity, length, and type of mistreatment should be developed and employed. Furthermore, we could not reject the possibility that the increase in workplace mistreatment and associated health problems could result from a continuous increase in the rate of workplace mistreatment as a trend, rather than the impact of COVID-19. Future data and analysis are required to fully discern the link between specific workplace mistreatment and health conditions and how the COVID-19 pandemic impacts this.

Nevertheless, a strength of our study is the relatively large sample size, the results of which can be generalized to the adult working population in South Korea. The results also provide an in-depth view of the association between workplace mistreatment and employee health before and after the COVID-19 pandemic. These findings have substantial implications regarding the regulation of work hours and emphasize the need to develop integrated working environment policies to eliminate harassment and discrimination not only among vulnerable working groups but all employees in general. Additionally, our results provide a considerable basis for future studies, which should focus on changes in the working environment due to COVID-19.

## 5. Conclusions

In conclusion, workplace mistreatment is associated with health problems and differently affects vulnerable groups. This association was found to have a stronger effect during the COVID-19 pandemic, supporting the initial hypothesis. Thus, efforts such as widespread enforcement of anti-harassment and anti-discrimination policies, establishing harassment and discrimination reporting infrastructure, and emphasis on employee well-being are needed, especially during the COVID-19 pandemic. In designing strategies to reduce health problems related to workplace mistreatment, particular attention should be paid to the vulnerable groups indicated in our study.

## Figures and Tables

**Figure 1 ijerph-19-12992-f001:**
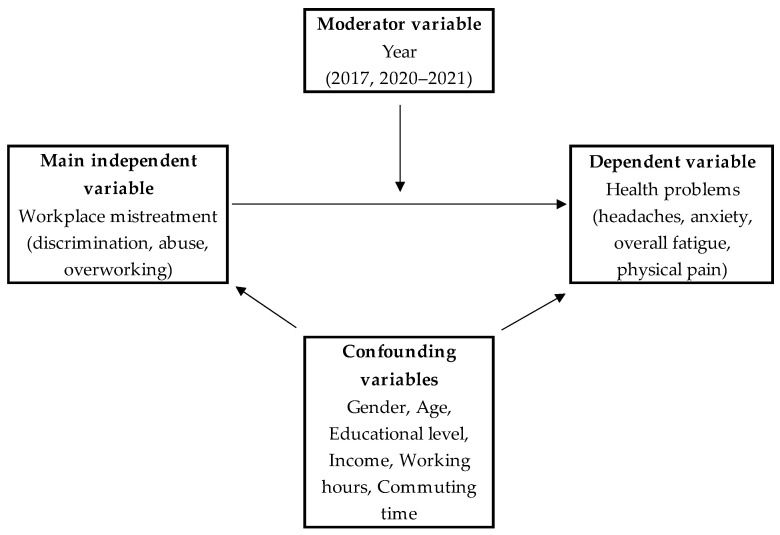
The theoretical model of the employed variables.

**Figure 2 ijerph-19-12992-f002:**
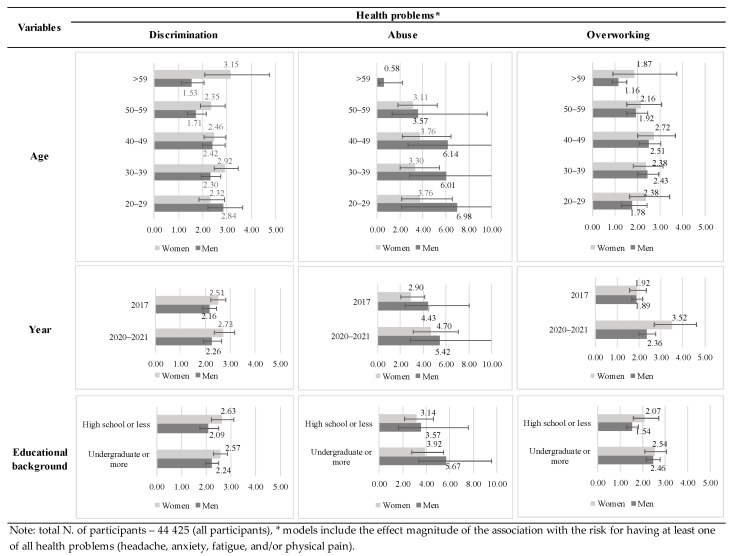
The result of the subgroup analysis was stratified by independent variables (age, year, and educational background).

**Table 1 ijerph-19-12992-t001:** General characteristics of the study population.

Variables	Health Problems											
	Before COVID-19 (2017)						During COVID-19 (2020–2021)					
	Total		Yes		No		Total		Yes		No	
	N	%	N	%	N	%	N	%	N	%	N	%
**(N = 44,425)**	21,460	100.0	7728	36.0	13,732	64.0	22,965	100.0	11,069	48.2	11,896	51.8
**Workplace mistreatment**												
No	17,540	81.7	5716	32.6	11,824	67.4	20,194	87.9	9189	45.5	11,005	54.5
Yes	3920	18.3	2012	51.3	1908	48.7	2771	12.1	1880	67.8	891	32.2
**Discrimination**												
No	18,869	87.9	6371	33.8	12,498	66.2	21,228	92.4	9887	46.6	11,341	53.4
Yes	2591	12.1	1357	52.4	1234	47.6	1737	7.6	1182	68.0	555	32.0
**Abuse**												
No	21,272	99.1	7606	35.8	13,666	64.2	22,724	99.0	10,870	47.8	11,854	52.2
Yes	188	0.9	122	64.9	66	35.1	241	1.0	199	82.6	42	17.4
**Overworking**												
No	19,953	93.0	6944	34.8	13,009	65.2	21,866	95.2	10,300	47.1	11,566	52.9
Yes	1507	7.0	784	52.0	723	48.0	1099	4.8	769	70.0	330	30.0
**Gender**												
Men	10,878	50.7	3700	34.0	7178	66.0	11,256	49.0	4981	44.3	6275	55.7
Women	10,582	49.3	4028	38.1	6554	61.9	11,709	51.0	6088	52.0	5621	48.0
**Age**												
>59	1614	7.5	808	50.1	806	49.9	2612	11.4	1636	62.6	976	37.4
50–59	4521	21.1	2028	44.9	2493	55.1	5460	23.8	2988	54.7	2472	45.3
40–49	6213	29.0	2269	36.5	3944	63.5	6230	27.1	3024	48.5	3206	51.5
30–39	5812	27.1	1765	30.4	4047	69.6	5794	25.2	2439	42.1	3355	57.9
20–29	3300	15.4	858	26.0	2442	74.0	2869	12.5	982	34.2	1887	65.8
**Educational level**												
High school or less	8144	37.9	3727	45.8	4417	54.2	8515	37.1	4788	56.2	3727	43.8
Undergraduate or more	13,316	62.1	4001	30.0	9315	70.0	14,450	62.9	6281	43.5	8169	56.5
**Income amount per month**												
<2,000,000 won	8673	40.4	3428	39.5	5245	60.5	6528	28.4	3419	52.4	3109	47.6
2,000,000–3,000,000 won	7077	33.0	2346	33.1	4731	66.9	8983	39.1	4197	46.7	4786	53.3
>3,000,000 won	5710	26.6	1954	34.2	3756	65.8	7454	32.5	3453	46.3	4001	53.7
**Working hours per week**												
>40 h	9477	44.2	4063	42.9	5414	57.1	13,017	56.7	9815	75.4	3202	24.6
40 h	10,689	49.8	3154	29.5	7535	70.5	13,342	58.1	5849	43.8	7493	56.2
<40 h	1294	6.0	511	39.5	783	60.5	2606	11.3	1405	53.9	1201	46.1
**Commuting time**												
>60 min	2586	12.1	908	35.1	1678	64.9	3705	16.1	1993	53.8	1712	46.2
30–60 min	9240	43.1	3 329	36.0	5911	64.0	9353	40.7	4555	48.7	4798	51.3
<30 min	9634	44.9	3 491	36.2	6143	63.8	9907	43.1	4521	45.6	5386	54.4

**Table 2 ijerph-19-12992-t002:** Results of factors associated with having health problems in 2017 and 2020–2021.

Mistreatment Type	Men								Women							
	Headaches		Anxiety		Overall Fatigue		Physical Pain		Headaches		Anxiety		Overall Fatigue		Physical Pain	
	OR *	95% CI	OR *	95% CI	OR *	95% CI	OR *	95% CI	OR *	95% CI	OR *	95% CI	OR *	95% CI	OR *	95% CI
**Discrimination**																
No	1.00		1.00		1.00		1.00		1.00		1.00		1.00		1.00	
Yes	2.40	(2.16–2.67)	2.83	(2.09–3.83)	1.29	(1.13–1.48)	0.95	(0.81–1.11)	2.51	(2.27–2.77)	2.38	(1.74–3.26)	1.30	(1.14–1.47)	1.20	(1.05–1.36)
**Abuse**																
No	1.00		1.00		1.00		1.00		1.00		1.00		1.00		1.00	
Yes	3.38	(2.35–4.85)	8.28	(4.50–15.23)	1.26	(0.76–2.08)	1.01	(0.56–1.83)	3.06	(2.42–3.85)	4.29	(2.42–7.60)	1.18	(0.86–1.64)	1.11	(0.81–1.53)
**Overworking**																
No	1.00		1.00		1.00		1.00		1.00		1.00		1.00		1.00	
Yes	2.40	(2.15–2.68)	2.45	(1.77–3.37)	1.10	(0.95–1.27)	0.98	(0.83–1.15)	2.15	(1.83–2.52)	3.02	(1.96–4.65)	1.17	(0.95–1.45)	1.29	(1.05–1.60)

Note: total N. of participants—44,425 (all participants), * adjusted to age, year, educational level, income amount/month, working hours/week, commuting time.

**Table 3 ijerph-19-12992-t003:** The results of subgroup analysis of the association between sexual harassment and health problems in women.

Variables	Health Problems			
	Before COVID-19 (2017)		During COVID-19 (2020–2021)	
	OR *	95% CI	OR *	95% CI
Abuse based on:				
Sexual harassment	2.09	(1.29–3.40)	2.59	(1.24–5.39)

Note: total N. of participants—22,291 (only women), * adjusted to age, educational level, income amount/month, working hours/week, commuting time.

**Table 4 ijerph-19-12992-t004:** The results of interaction factors associated with health problems.

Variables	Health Problems					
	OR	95% CI	*p*-Value	OR *	95% CI	*p*-Value
**Year**						
2017	1.00			1.00		
2020–2021	1.36	(1.33–1.40)	<0.0001	1.42	(1.38–1.46)	<0.0001
**Mistreatment**						
No	1.00			1.00		
Yes	1.53	(1.49–1.57)	<0.0001	1.47	(1.42–1.51)	<0.0001
**Year x Mistreatment**						
2017 × No mistreatment	1.00			1.00		
2020 × Mistreatment	1.04	(1.01–1.07)	0.0086	1.03	(1.00–1.06)	0.0321

Note: total n. of participants—44,425 (all participants), * adjusted to age, year, educational level, income amount/month, working hours/week, commuting time.

## Data Availability

The Korean version of the Working Conditions Survey (KWCS) is publicly available at http://www.kosha.or.kr/ (accessed on 23 April 2022) under permission from Korea Occupational Safety and Health Agency (KOSHA).

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
