# Peer review of "Workplace Mistreatment and Health Conditions Prior and during the COVID-19 in South Korea: A Cross-Sectional Study"

_ijerph, 2022, doi:10.3390/ijerph192012992_

Round 1
Reviewer 1 Report
Dear authors
please supplement your introduction with additional theoretical background, especially regarding previous findings on impact of working conditions on workers' health. Only later in the manuscript you mention some references (31 and 32) about that but your introductory theoretical framework should support your research aim and your notion of existing knowledge gap on the topic.
Additionally, you should end your introduction with clearly specifying your hypothesis generated from the research aim. In the end of the paper, your discussion and/or conclusion sections should state whether those hypothesis have been confirmed or not.
Your suggestion in line 232 that those with higher education find mistreatment more distressing than those with lower education should be substantiated, otherwise it remains speculation rather than a scientific claim.
Author Response
We were pleased to have the opportunity to revise our paper. In revising our paper, we have carefully considered your comments and suggestions. As instructed, we have attempted to explain the changes made in reaction to all of the reviewers’ comments. The reviewers’ comments were very helpful overall, and we appreciate the constructive feedback on our original submission. After addressing the issues raised, we feel the quality of the paper has greatly improved, and we hope you agree. Our response to each comment is as follows, and we attach a revision note with the highlighted, revised sections of the manuscript. Again, thank you for the valuable and helpful comments.

Reviewer 2 Report
This is a very interesting study that aimed to examine the association between different types of workplace mistreatment and health problems among full-time workers in South Korea, comparing effect estimates prior to and during the COVID-19 pandemic. It is a well written manuscript. The text is clear and terse. Important information regarding the background is well described in the introduction. Methods and results are adequately described, in general. The study has a few limitation regarding its design and measurements, but they are adequately discussed in the discussion section. The manuscript may benefit from very few clarifications, so a few considerations are made below for each section.
Methods: It may be interesting to specify a clear theoretical model and the relationship between all relevant variables for the main analyses. The authors cited several variables, but it nos not clear which one is a possible confounder (or mediator, or effect modifier).
Results / Tables: I would suggest to include in tables footnotes (tables 2 to 4) what variables were included in the full models. It would be useful to specify which variables corresponded to "all confounders" in table 2 and to "other variables" in table 4, as it is not specified in the methods . Also, it would be important to describe in tables titles (or somewhere in the table) the number of participants included in each analysis. Were all participants included? Did the authors have full information for all participants?
Table 2 is quite large, which may not be recommended for a better reading and understanding. Maybe the crude model (model 1) could be excluded from the table, showing only models 2 and 3.
I found the sentence "Table 3 shows the results of a subgroup analysis stratified by the association between sexual harassment and health problems in women" a bit confusing. What was the variable that authors stratified? Was it gender or year? Why did the authors presented only results for women, but not for men. Men are usually less exposed to sexual harassment than women, but this type of violence may also occur in men and have serious effects on their health.
In the sentence "compared to 2017, in 2020-2021 a significant increase in the likelihood of women having health problems related to sexual harassment was found (2017: OR 2.09, 95% CI 1.29–3.40; 2020: OR 2.59, 95% CI 1.24–5.39)." What did the authors mean with "significant increase"? Despite a higher OR in 2020 compared to 2017, confidence intervals are similar.
It may be preferable to discuss the effects in terms of their magnitude rather than using expressions such as "one model was more significant than other". It might not be the most appropriate form to describe results, once p-values are related to each specified hypotheses test.
Minor comments:
Please revise the term "One study7" in page 2 (line 53).
Author Response

(The authors gave the same response as above.)

Reviewer 3 Report
This manuscript describes the association of discrimination, abuse, and overworking with health problems in South Korean full-time employees. Two time periods are compared: pre corona and during corona. For a better classification of the analyzed corona period, it would be good to know whether the survey took place during a period with high incidence and high mortality or during two waves. Furthermore, the observation period from October 2020 to April 2021 is very long. Was the participation of the subjects equally distributed over time?
Assuming that South Korea also experienced increased home office use during the pandemic, it is perhaps not surprising that fewer employees experienced mistreatment. On the other hand, there was an increased risk of sexual harassment among women during COVID-19, which has been excellently discussed by the authors. However, I am wondering how based on the question "In the past 12 months, have you been subjected to any of the following in the course of your work: physical violence, sexual harassment, bullying/harassment?" you can distinguish whether it is sexual harassment?
Furthermore, it was not made clear to me whether the same individuals participated at both time points. In addition, was there also data on different industries or occupational groups that could be analyzed? The authors discuss the fact that healthcare workers in particular are confronted with an increased workload during the pandemic (line 269). Thus, the observed risks could differ between different occupational groups.
The manuscript findings are certainly interesting, but I would like to see the authors address the issues raised. Some more detailed comments below:
Introducation:
- Typo in line 53: „One study7“
- Lines 58/59: I doubt that workplace violence is the most dangerous occupational risk factor leading to health problems in the work environment. Is this actually more dangerous (and more often) than occupational exposures to hazardous materials? I would encourage the authors to review this statement and provide more details if necessary.
Methods:
- 2.1. Data source and sample: Were the same people always interviewed in the fifth and sixth survey, so that multiple measurements would have to be taken into account in the statistical analysis?
- Line 135: Is 40 hours per week the regular working week in Korea? Was this the reason for the cut-off? Are employees working less than 40 hours per week also considered full-time employees (Table 1)? Should they not be excluded from the analysis if only full-time employees were to be surveyed?
- The terms sex (line 140) and gender (line 148) are apparently used synonymously. This is not correct and should be revised throughout the whole manuscript.
-Line 141: Why was no upper age limit chosen? What is the official retirement age in South Korea?
Results:
- Table 1: Perhaps "workplace mistreatment" could also be added as row to Table 1. Otherwise, it is not possible to understand the figures from lines 160-164 in the manuscript.
- Table 2: „All confounders“ should be named in the table note. Furthermore, Table 2 is very big with many (too many ?) numbers given. For the main body of the paper, could it be better to present only one model (Model 1, Model 2, or Model 3) and the other models in the appendix, since the observed associations were consistent in all models as stated.
- Figure 1: It should be clarified that Figure 1 shows the results of the models with the risk for any health problem. Thus, the differences to Table 1 becomes clear at first glance. In general, though, it would be interesting to see what the unstratified estimators for any health problem look like? Is the association stronger for men or women?
Discussion
- Typo in line 219: „type of workplace mistreatment type“
- Third paragraph: According to the authors, the most vulnerable groups are the oldest and youngest employees. How do you explain the difference between the sexes? This should be discussed in more detail.
References: The list of references is very long. A few references (especially some of the very old pre-millennial ones) could certainly be done without.
Author Response

(The authors gave the same response as above.)
